# Conditional Generative Moment-Matching Networks

**Yong Ren,  Jialian Li,  Yucen Luo,  Jun Zhu**[*]
Dept. of Comp. Sci. & Tech., TNList Lab; Center for Bio-Inspired Computing Research
State Key Lab for Intell. Tech. & Systems, Tsinghua University, Beijing, China
{renyong15, luoyc15, jl12}@mails.tsinghua.edu.cn; dcszj@tsinghua.edu.cn

## Abstract

Maximum mean discrepancy (MMD) has been successfully applied to learn deep generative models for characterizing a joint distribution of variables via kernel mean embedding. In this paper, we present conditional generative moment-matching networks (CGMMN), which learn a conditional distribution given some input variables based on a conditional maximum mean discrepancy (CMMD) criterion. The learning is performed by stochastic gradient descent with the gradient calculated by back-propagation. We evaluate CGMMN on a wide range of tasks, including predictive modeling, contextual generation, and Bayesian dark knowledge, which distills knowledge from a Bayesian model by learning a relatively small CGMMN student network. Our results demonstrate competitive performance in all the tasks.

## 1  Introduction

Deep generative models (DGMs) characterize the distribution of observations with a multilayered structure of hidden variables under nonlinear transformations. Among various deep learning methods, DGMs are natural choice for those tasks that require probabilistic reasoning and uncertainty estimation, such as image generation [1], multimodal learning [30], and missing data imputation. Recently, the predictive power, which was often shown inferior to pure recognition networks (e.g., deep convolutional networks), has also been significantly improved by employing the discriminative max-margin learning [18].

For the arguably more challenging unsupervised learning, [5] presents a generative adversarial network (GAN), which adopts a game-theoretical min-max optimization formalism. GAN has been extended with success in various tasks [21, 1]. However, the min-max formalism is often hard to solve. The recent work [19, 3] presents generative moment matching networks (GMMN), which has a simpler objective function than GAN while retaining the advantages of deep learning. GMMN defines a generative model by sampling from some simple distribution (e.g., uniform) followed through a parametric deep network. To learn the parameters, GMMN adopts maximum mean discrepancy (MMD) [7], a moment matching criterion where kernel mean embedding techniques are used to avoid unnecessary assumptions of the distributions. Back-propagation can be used to calculate the gradient as long as the kernel function is smooth.

A GMMN network estimates the joint distribution of a set of variables. However, we are more interested in a conditional distribution in many cases, including (1) *predictive modeling*: compared to a generative model that defines the joint distribution $p(\boldsymbol{x}, \boldsymbol{y})$ of input data $\boldsymbol{x}$ and response variable $\boldsymbol{y}$, a conditional model $p(\boldsymbol{y}|\boldsymbol{x})$ is often more direct without unnecessary assumptions on modeling $\boldsymbol{x}$, and leads to better performance with fewer training examples [23, 16]; (2) *contextual generation*: in some cases, we are interested in generating samples based on some context, such as class labels [21], visual attributes [32] or the input information in cross-modal generation (e.g., from image to text [31]

---

[*]Corresponding author

or vice versa [2]); and (3) *building large networks*: conditional distributions are essential building blocks of a large generative probabilistic model. One recent relevant work [1] provides a good example of stacking multiple conditional GAN networks [21] in a Laplacian pyramid structure to generate natural images.

In this paper, we present conditional generative moment-matching networks (CGMMN) to learn a flexible conditional distribution when some input variables are given. CGMMN largely extends the capability of GMMN to address a wide range of application problems as mentioned above, while keeping the training process simple. Specifically, CGMMN admits a simple generative process, which draws a sample from a simple distribution and then passes the sample as well as the given conditional variables through a deep network to generate a target sample. To learn the parameters, we develop conditional maximum mean discrepancy (CMMD), which measures the Hilbert-Schmidt norm (generalized Frobenius norm) between the kernel mean embedding of an empirical conditional distribution and that of our generative model. Thanks to the simplicity of the conditional generative model, we can easily draw a set of samples to estimate the kernel mean embedding as well as the CMMD objective. Then, optimizing the objective can be efficiently implemented via back-propagation. We evaluate CGMMN in a wide range of tasks, including predictive modeling, contextual generation, and Bayesian dark knowledge [15], an interesting case of distilling dark knowledge from Bayesian models. Our results on various datasets demonstrate that CGMMN can obtain competitive performance in all these tasks.

## 2 Preliminary

In this section, we briefly review some preliminary knowledge, including maximum mean discrepancy (MMD) and kernel embedding of conditional distributions.

### 2.1 Hilbert Space Embedding

We begin by providing an overview of Hilbert space embedding, where we represent distributions by elements in a *reproducing kernel Hilbert space* (RKHS). A RKHS $\mathcal{F}$ on $\mathcal{X}$ with kernel $k$ is a Hilbert space of functions $f : \mathcal{X} \to \mathbb{R}$. Its inner product $\langle \cdot, \cdot \rangle_{\mathcal{F}}$ satisfies the reproducing property: $\langle f(\cdot), k(\boldsymbol{x}, \cdot) \rangle_{\mathcal{F}} = f(\boldsymbol{x})$. Kernel functions are not restricted on $\mathbb{R}^d$. They can also be defined on graphs, time series and structured objects [11]. We usually view $\phi(\boldsymbol{x}) := k(\boldsymbol{x}, \cdot)$ as a (usually infinite dimension) feature map of $\boldsymbol{x}$. The most interesting part is that we can embed a distribution by taking expectation on its feature map:

$$\mu_X := \mathbb{E}_X[\phi(X)] = \int_{\Omega} \phi(X) dP(X).$$

If $\mathbb{E}_X[k(X, X)] \leq \infty$, $\mu_X$ is guaranteed to be an element in the RKHS. This kind of kernel mean embedding provides us another perspective on manipulating distributions whose parametric forms are not assumed, as long as we can draw samples from them. This technique has been widely applied in many tasks, including feature extractor, density estimation and two-sample test [27, 7].

### 2.2 Maximum Mean Discrepancy

Let $X = \{\boldsymbol{x}_i\}_{i=1}^{N}$ and $Y = \{\boldsymbol{y}_j\}_{j=1}^{M}$ be the sets of samples from distributions $P_X$ and $P_Y$, respectively. Maximum Mean Discrepancy (MMD), also known as kernel two sample test [7], is a frequentist estimator to answer the query whether $P_X = P_Y$ based on the observed samples. The basic idea behind MMD is that if the generating distributions are identical, all the statistics are the same. Formally, MMD defines the following difference measure:

$$\text{MMD}[\mathcal{K}, P_X, P_Y] := \sup_{f \in \mathcal{K}} (\mathbb{E}_X[f(X)] - \mathbb{E}_Y[f(Y)]),$$

where $\mathcal{K}$ is a class of functions. [7] found that the class of functions in a universal RKHS $\mathcal{F}$ is rich enough to distinguish any two distributions and MMD can be expressed as the difference of their mean embeddings. Here, universality requires that $k(\cdot, \cdot)$ is continuous and $\mathcal{F}$ is dense in $C(X)$ with respect to the $L_{\infty}$ norm, where $C(X)$ is the space of bounded continuous functions on $X$. We summarize the result in the following theorem:

**Theorem 1** *[7] Let $\mathcal{K}$ be a unit ball in a universal RKHS $\mathcal{F}$, defined on the compact metric space $\mathcal{X}$, with an associated continuous kernel $k(\cdot, \cdot)$. When the mean embedding $\mu_p, \mu_q \in \mathcal{F}$, the MMD objective function can be expressed as $MMD[\mathcal{K}, p, q] = \|\mu_p - \mu_q\|_{\mathcal{F}}^2$. Besides, $MMD[\mathcal{K}, p, q] = 0$ if and only if $p = q$.*

In practice, an estimate of the MMD objective compares the square difference between the empirical kernel mean embeddings:

$$\widehat{\mathcal{L}}_{\mathrm{MMD}}^2 = \left\| \frac{1}{N} \sum_{i=1}^{N} \phi(\boldsymbol{x}_i) - \frac{1}{M} \sum_{j=1}^{M} \phi(\boldsymbol{y}_i) \right\|_{\mathcal{F}}^2,$$

which can be easily evaluated by expanding the square and using the associated kernel $k(\cdot, \cdot)$. Asymptotically, $\widehat{\mathcal{L}}_{\mathrm{MMD}}^2$ is an unbiased estimator.

### 2.3 Kernel Embedding of Conditional Distributions

The kernel embedding of a conditional distribution $P(Y|X)$ is defined as: $\mu_{Y|\boldsymbol{x}} := \mathbb{E}_{Y|\boldsymbol{x}}[\phi(Y)] = \int_{\Omega} \phi(\boldsymbol{y}) dP(\boldsymbol{y}|\boldsymbol{x})$. Unlike the embedding of a single distribution, the embedding of a conditional distribution is not a single element in RKHS, but sweeps out a family of points in the RKHS, each indexed by a fixed value of $\boldsymbol{x}$. Formally, the embedding of a conditional distribution is represented as an operator $C_{Y|X}$, which satisfies the following properties:

$$1. \ \mu_{Y|\boldsymbol{x}} = C_{Y|X}\phi(\boldsymbol{x}); \quad 2. \ \mathbb{E}_{Y|\boldsymbol{x}}[g(Y)|\boldsymbol{x}] = \langle g, \mu_{Y|\boldsymbol{x}} \rangle_{\mathcal{G}}, \tag{1}$$

where $\mathcal{G}$ is the RKHS corresponding to $Y$.

[29] found that such an operator exists under some assumptions, using the technique of cross-covariance operator $C_{XY} : \mathcal{G} \to \mathcal{F}$:

$$C_{XY} := \mathbb{E}_{XY}[\phi(X) \otimes \phi(Y)] - \mu_X \otimes \mu_Y,$$

where $\otimes$ is the tensor product. An interesting property is that $C_{XY}$ can also be viewed as an element in the tensor product space $\mathcal{G} \otimes \mathcal{F}$. The result is summarized as follows.

**Theorem 2** *[29] Assuming that $\mathbb{E}_{Y|X}[g(Y)|X] \in \mathcal{F}$, the embedding of conditional distributions $C_{Y|X}$ defined as $C_{Y|X} := \mathcal{C}_{YX}\mathcal{C}_{XX}^{-1}$ satisfies properties 1 and 2.*

Given a dataset $\mathcal{D}_{XY} = \{(\boldsymbol{x}_i, \boldsymbol{y}_i)\}_{i=1}^{N}$ of size $N$ drawn *i.i.d.* from $P(X, Y)$, we can estimate the conditional embedding operator as $\widehat{C}_{Y|X} = \Phi(K + \lambda I)^{-1}\Upsilon^{\top}$, where $\Phi = (\phi(\boldsymbol{y}_1), ..., \phi(\boldsymbol{y}_N))$, $\Upsilon = (\phi(\boldsymbol{x}_1), ..., \phi(\boldsymbol{x}_N))$, $K = \Upsilon^{\top}\Upsilon$ and $\lambda$ serves as regularization. The estimator is an element in the tensor product space $\mathcal{F} \otimes \mathcal{G}$ and satisfies properties 1 and 2 asymptotically. When the domain of $X$ is finite, we can also estimate $C_{XX}^{-1}$ and $C_{YX}$ directly (See Appendix A.2.2 for more details).

## 3 Conditional Generative Moment-Matching Networks

We now present CGMMN, including a conditional maximum mean discrepancy criterion as the training objective, a deep generative architecture and a learning algorithm.

### 3.1 Conditional Maximum Mean Discrepancy

Given conditional distributions $P_{Y|X}$ and $P_{Z|X}$, we aim to test whether they are the same in the sense that when $X = \boldsymbol{x}$ is fixed whether $P_{Y|\boldsymbol{x}} = P_{Z|\boldsymbol{x}}$ holds or not. When the domain of $X$ is finite, a straightforward solution is to test whether $P_{Y|\boldsymbol{x}} = P_{Z|\boldsymbol{x}}$ for each $\boldsymbol{x}$ separately by using MMD. However, this is impossible when $X$ is continuous. Even in the finite case, as the separate tests do not share statistics, we may need an extremely large number of training data to test a different model for each single value of $\boldsymbol{x}$. Below, we present a conditional maximum mean discrepancy criterion, which avoids the above issues.

Recall the definition of kernel mean embedding of conditional distributions. When $X = \boldsymbol{x}$ is fixed, we have the kernel mean embedding $\mu_{Y|\boldsymbol{x}} = C_{Y|X}\phi(\boldsymbol{x})$. As a result, if we have $C_{Y|X} = C_{Z|X}$, then $\mu_{Y|\boldsymbol{x}} = \mu_{Z|\boldsymbol{x}}$ is also satisfied for every fixed $\boldsymbol{x}$. By the virtue of Theorem 1, that $P_{Y|\boldsymbol{x}} = P_{Z|\boldsymbol{x}}$ follows as the following theorem states.

**Theorem 3** *Assuming that $\mathcal{F}$ is a universal RKHS with an associated kernel $k(\cdot, \cdot)$, $\mathbb{E}_{Y|X}[g(Y)|X] \in \mathcal{F}$, $\mathbb{E}_{Z|X}[g(Z)|X] \in \mathcal{F}$ and $C_{Y|X}, C_{Z|X} \in \mathcal{F} \otimes \mathcal{G}$. If the embedding of conditional distributions $C_{Y|X} = C_{Z|X}$, then $P_{Y|X} = P_{Z|X}$ in the sense that for every fixed $\boldsymbol{x}$, we have $P_{Y|\boldsymbol{x}} = P_{Z|\boldsymbol{x}}$.*

The above theorem gives us a sufficient condition to guarantee that two conditional distributions are the same. We use the operators to measure the difference of two conditional distributions and we call it *conditional maximum mean discrepancy* (CMMD), which is defined as follows:

$$\mathcal{L}^2_{\text{CMMD}} = \left\| C_{Y|X} - C_{Z|X} \right\|^2_{\mathcal{F} \otimes \mathcal{G}}.$$

Suppose we have two sample sets $\mathcal{D}^s_{XY} = \{(\boldsymbol{x}_i, \boldsymbol{y}_i)\}^N_{i=1}$ and $\mathcal{D}^d_{XY} = \{(\boldsymbol{x}_i, \boldsymbol{y}_i)\}^M_{i=1}$. Similar as in MMD, in practice we compare the square difference between the empirical estimates of the conditional embedding operators:

$$\widehat{\mathcal{L}}^2_{\text{CMMD}} = \left\| \widehat{C}^d_{Y|X} - \widehat{C}^s_{Y|X} \right\|^2_{\mathcal{F} \otimes \mathcal{G}},$$

where the superscripts $s$ and $d$ denote the two sets of samples, respectively. For notation clarity, we define $\widetilde{K} = K + \lambda I$. Then, using kernel tricks, we can compute the difference only in term of kernel gram matrices:

$$
\begin{aligned}
\widehat{\mathcal{L}}^2_{\text{CMMD}} &= \left\| \Phi_d (K_d + \lambda I)^{-1} \Upsilon^\top_d - \Phi_s (K_s + \lambda I)^{-1} \Upsilon^\top_s \right\|^2_{\mathcal{F} \otimes \mathcal{G}} \\
&= \text{Tr}\left( K_d \widetilde{K}^{-1}_d L_d \widetilde{K}^{-1}_d \right) + \text{Tr}\left( K_s \widetilde{K}^{-1}_s L_s \widetilde{K}^{-1}_s \right) - 2 \cdot \text{Tr}\left( K_{sd} \widetilde{K}^{-1}_d L_{ds} \widetilde{K}^{-1}_s \right),
\end{aligned}
\tag{2}
$$

where $\Phi_d := (\phi(\boldsymbol{y}^d_1), ..., \phi(\boldsymbol{y}^d_N))$ and $\Upsilon_d := (\phi(\boldsymbol{x}^d_1), ..., \phi(\boldsymbol{x}^d_N))$ are implicitly formed feature matrices, $\Phi_s$ and $\Upsilon_s$ are defined similarly for dataset $\mathcal{D}^s_{XY}$. $K_d = \Upsilon^\top_d \Upsilon_d$ and $K_s = \Upsilon^\top_s \Upsilon_s$ are the gram matrices for input variables, while $L_d = \Phi^\top_d \Phi_d$ and $L_s = \Phi^\top_s \Phi_s$ are the gram matrices for output variables. Finally, $K_{sd} = \Upsilon^\top_s \Upsilon_d$ and $L_{ds} = \Phi^\top_d \Phi_s$ are the gram matrices between the two datasets on input and out variables, respectively.

It is worth mentioning that we have assumed that the conditional mean embedding operator $C_{Y|X} \in \mathcal{F} \otimes \mathcal{G}$ to have the CMMD objective well-defined, which needs some smoothness assumptions such that $C^{-3/2}_{XX} C_{XY}$ is Hilbert-Schmidt [8]. In practice, the assumptions may not hold, however, the empirical estimator $\Phi(K + \lambda I)^{-1} \Upsilon^\top$ is always an element in the tensor product space which gives as a well-justified approximation (i.e., the Hilbert-Schmidt norm exists) for practical use [29].

**Remark 1** *Taking a close look on the objectives of MMD and CMMD, we can find some interesting connections. Suppose $N = M$. By omitting the constant scalar, the objective function of MMD can be rewritten as*

$$\widehat{\mathcal{L}}^2_{MMD} = \text{Tr}(L_d \cdot \mathbf{1}) + \text{Tr}(L_s \cdot \mathbf{1}) - 2 \cdot \text{Tr}(L_{ds} \cdot \mathbf{1}),$$

*where $\mathbf{1}$ is the matrix with all entities equaling to $1$. The objective function of CMMD can be expressed as*

$$\widehat{\mathcal{L}}^2_{CMMD} = \text{Tr}(L_d \cdot C_1) + \text{Tr}(L_s \cdot C_2) - 2 \cdot \text{Tr}(L_{ds} \cdot C_3),$$

*where $C_1, C_2, C_3$ are some matrices based on the conditional variables $\boldsymbol{x}$ in both data sets. The difference is that instead of putting uniform weights on the gram matrix as in MMD, CMMD applies non-uniform weights, reflecting the influence of conditional variables. Similar observations have been shown in [29] for the conditional mean operator, where the estimated conditional embedding $\mu_{Y|\boldsymbol{x}}$ is a non-uniform weighted combination of $\phi(\boldsymbol{x}_i)$.*

## 3.2 CGMMN Nets

We now present a conditional DGM and train it by the CMMD criterion. One desirable property of the DGM is that we can easily draw samples from it to estimate the CMMD objective. Below, we present such a network that takes both the given conditional variables and an extra set of random variables as inputs, and then passes through a deep neural network with nonlinear transformations to produce the samples of the target variables.

Specifically, our network is built on the fact that for any distribution $\mathcal{P}$ on sample space $\mathbb{K}$ and any continuous distribution $Q$ on $\mathbb{L}$ that are regular enough, there is a function $G : \mathbb{L} \to \mathbb{K}$ such that $G(\boldsymbol{x}) \sim \mathcal{P}$, where $\boldsymbol{x} \sim \mathcal{Q}$ [12]. This fact has been recently explored by [3, 19] to define a deep generative model and estimate the parameters by the MMD criterion. For a conditional model, we would like the function $G$ to depend on the given values of input variables. This can be fulfilled via a process as illustrated in Fig. 1, where the inputs of a deep neural network (DNN) consist of two parts — the input variables $\boldsymbol{x}$ and an extra set of stochastic variables $H \in \mathbb{R}^d$ that follow some continuous distribution. For simplicity, we put a uniform prior on each hidden unit $p(\boldsymbol{h}) = \prod_{i=1}^{d} U(h_i)$, where $U(h) = \boldsymbol{I}_{(0 \leq h \leq 1)}$ is a uniform distribution on $[0, 1]$ and $\boldsymbol{I}_{(\cdot)}$ is the indicator

function that equals to 1 if the predicate holds and 0 otherwise. After passing both $x$ and $h$ through the DNN, we get a sample from the conditional distribution $P(Y|x)$: $y = f(x, h|w)$, where $f$ denotes the deterministic mapping function represented by the network with parameters $w$. By default, we concatenate $x$ and $h$ and fill $\widetilde{x} = (x, h)$ into the network. In this case, we have $y = f(\widetilde{x}|w)$.

Due to the flexibility and rich capability of DNN on fitting nonlinear functions, this generative process can characterize various conditional distributions well. For example, a simple network can consist of multiple layer perceptrons (MLP) activated by some non-linear functions such as the rectified linear unit (ReLu) [22]. Of course the hidden layer is not restricted to MLP, as long as it supports gradient propagation. We also use convolutional neural networks (CNN) as hidden layers [25] in our experiments. It is worth mentioning that there exist other ways to combine the conditional variables $x$ with the auxiliary variables $H$. For example, we can add a corruption noise to the conditional variables $x$ to produce the input of the network, e.g., define $\widetilde{x} = x + h$, where $h$ may follow a Gaussian distribution $\mathcal{N}(0, \eta I)$ in this case.

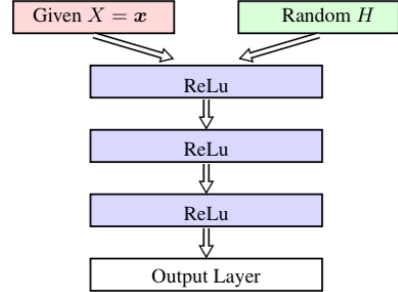

Figure 1: An example architecture of CGMMN networks.

With the above generative process, we can train the network by optimizing the CMMD objective with proper regularization. Specifically, let $\mathcal{D}_{XY}^s = \{(x_i^d, y_i^d)\}_{i=1}^N$ denote the given training dataset. To estimate the CMMD objective, we draw a set of samples from the above generative model, where the conditional variables can be set by sampling from the training set with/without small perturbation (More details are in the experimental section). Thanks to its simplicity, the sampling procedure can be easily performed. Precisely, we provide each $x$ in the training dataset to the generator to get a new sample and we denote $\mathcal{D}_{XY}^d = \{(x_i^s, y_i^s)\}_{i=1}^M$ as the generated samples. Then, we can optimize the CMMD objective in Eq. (2) by gradient descent. See more details in Appendix A.1.

---

**Algorithm 1** Stochastic gradient descent for CGMMN

---

1: **Input:** Dataset $\mathcal{D} = \{(x_i, y_i)\}_{i=1}^N$
2: **Output:** Learned parameters $w$
3: Randomly divide training dataset $\mathcal{D}$ into mini batches
4: **while** Stopping criterion not met **do**
5:     Draw a minibatch $\mathcal{B}$ from $\mathcal{D}$;
6:     For each $x \in \mathcal{B}$, generate a $y$; and set $\mathcal{B}'$ to contain all the generated $(x, y)$;
7:     Compute the gradient $\frac{\partial \widehat{\mathcal{L}}_{\text{CMMD}}^2}{\partial w}$ on $\mathcal{B}$ and $\mathcal{B}'$;
8:     Update $w$ using the gradient with proper regularizer.
9: **end while**

---

Note that the inverse matrices $\widetilde{K}_s^{-1}$ and $\widetilde{K}_d^{-1}$ in the CMMD objective are independent of the model parameters, suggesting that we are not restricted to use differentiable kernels on the conditional variables $x$. Since the computation cost for kernel gram matrix grows cubically with the sample size, we present an mini-batch version algorithm in Alg. 1 and some discussions can be found in Appendix A.2.1.

## 4 Experiments

We now present a diverse range of applications to evaluate our model, including predictive modeling, contextual generation and an interesting case of Bayesian dark knowledge [15]. Our results demonstrate that CGMMN is competitive in all the tasks.

### 4.1 Predictive Performance

#### 4.1.1 Results on MNIST dataset

We first present the prediction performance on the widely used MINIST dataset, which consists of images in 10 classes. Each image is of size $28 \times 28$ and the gray-scale is normalized to be in range $[0, 1]$. The whole dataset is divided into 3 parts with $50,000$ training examples, $10,000$ validation examples and $10,000$ testing examples.

For prediction task, the conditional variables are the images $x \in [0, 1]^{28 \times 28}$, and the generated sample is a class label, which is represented as a vector $y \in \mathbb{R}_+^{10}$ and each $y_i$ denotes the confidence that $x$ is in class $i$. We consider two types of architectures in CGMMN — MLP and CNN.

We compare our model, denoted as CGMMN in the MLP case and CGMMN-CNN in the CNN case, with Varitional Auto-encoder (VA) [14], which is an unsupervised DGM learnt by stochastic variational methods. To use VA for classification, a subsequent classifier is built — We first learn feature representations by VA and then learn a linear SVM on these features using Pegasos algorithm [26]. We also compare with max-margin DGMs (denoted as MMVA with MLP as hidden layers and CMMVA in the CNN case) [18], which is a state-of-the-art DGM for prediction, and several other strong baselines, including Stochastic Pooling [33], Network in Network [20], Maxout Network [6] and Deeply-supervised nets (DSN) [17].

Table 1: Error rates (%) on MNIST dataset

| Model | Error Rate |
|---|---|
| VA+Pegasos [18] | 1.04 |
| MMVA [18] | 0.90 |
| CGMMN | 0.97 |
| CVA + Pegasos [18] | 1.35 |
| CGMMN-CNN | 0.47 |
| Stochastic Pooling [33] | 0.47 |
| Network in Network [20] | 0.47 |
| Maxout Network [6] | 0.45 |
| CMMVA [18] | 0.45 |
| DSN [17] | 0.39 |

In the MLP case, the model architecture is shown in Fig. 1 with an uniform distribution for hidden variables of dimension 5. Note that since we do not need much randomness for the prediction task, this low-dimensional hidden space is sufficient. In fact, we did not observe much difference with a higher dimension (e.g., 20 or 50), which simply makes the training slower. The MLP has 3 hidden layers with hidden unit number $(500, 200, 100)$ with the ReLu activation function. A minibatch size of $500$ is adopted. In the CNN case, we use the same architecture as [18], where there are 32 feature maps in the first two convolutional layers and $64$ feature maps in the last three hidden layers. An MLP of $500$ hidden units is adopted at the end of convolutional layers. The ReLu activation function is used in the convoluational layers and sigmoid function in the last layer. We do not pre-train our model and a minibatch size of $500$ is adopted as well. The total number of parameters in the network is comparable with the competitors [18, 17, 20, 6].

In both settings, we use AdaM [13] to optimize parameters. After training, we simply draw a sample from our model conditioned on the input image and choose the index of maximum element of $\boldsymbol{y}$ as its prediction. Table 1 shows the results. We can see that CGMMN-CNN is competitive with various state-of-the-art competitors that do not use data augumentation or multiple model voting (e.g., CMMVA). DSN benefits from using more supervision signal in every hidden layer and outperforms the other competitors.

### 4.1.2 Results on SVHN dataset

We then report the prediction performance on the Street View House Numbers (SVHN) dataset. SVHN is a large dataset consisting of color images of size $32 \times 32$ in 10 classes. The dataset consists of $598,388$ training examples, $6,000$ validation examples and $26,032$ testing examples. The task is significantly harder than classifying hand-written digits. Following [25, 18], we preprocess the data by Local Contrast Normalization (LCN). The architecture of out network is similar to that in MNIST and we only use CNN as middle layers here. A minibatch size of $300$ is used and the other settings are the same as the MNIST experiments.

Table 2: Error rates (%) on SVHN dataset

| Model | Error Rate |
|---|---|
| CVA+Pegasos [18] | 25.3 |
| CGMMN-CNN | 3.13 |
| CNN [25] | 4.9 |
| CMMVA [18] | 3.09 |
| Stochastic Pooling [33] | 2.80 |
| Network in Network [20] | 2.47 |
| Maxout Network [6] | 2.35 |
| DSN [17] | 1.92 |

Table 2 shows the results. Through there is a gap between our CGMMN and some discriminative deep networks such as DSN, our results are comparable with those of CMMVA, which is the state-of-the-art DGM for prediction. CGMMN is compatible with various network architectures and we are expected to get better results with more sophisticated structures.

## 4.2 Generative Performance
### 4.2.1 Results on MNIST dataset

We first test the generating performance on the widely used MNIST dataset. For generating task, the conditional variables are the image labels. Since $\boldsymbol{y}$ takes a finite number of values, as mentioned in Sec. 2.3, we estimate $C_{YX}$ and $C_{XX}^{-1}$ directly and combine them as the estimation of $C_{Y|X}$ (See Appendix A.2.2 for practical details).

The architecture is the same as before but exchanging the position of $\boldsymbol{x}$ and $\boldsymbol{y}$. For the input layer, besides the label information $\boldsymbol{y}$ as conditional variables (represented by a one-hot-spot vector of dimension 10), we further draw a sample from a uniform distribution of dimension 20, which is

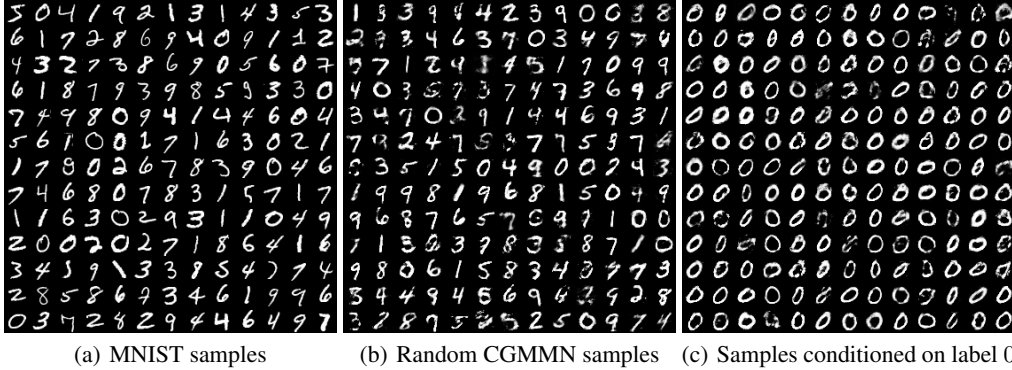

| (a) MNIST samples | (b) Random CGMMN samples | (c) Samples conditioned on label 0 |

Figure 2: Samples in (a) are from MNIST dataset; (b) are generated randomly from our CGMMN network; (c) are generated randomly from CGMMN with conditions on label $y = 0$. Both (b) and (c) are generated after running $500$ epoches.

sufficiently large. Overall, the network is a 5-layer MLP with input dimension 30 and the middle layer hidden unit number $(64, 256, 256, 512)$, and the output layer is of dimension $28 \times 28$, which represents the image in pixel. A minibatch of size 200 is adopted.

Fig. 2 shows some samples generated using our CGMMN, where in (b) the conditional variable $\boldsymbol{y}$ is randomly chosen from the 10 possible values, and in (c) $\boldsymbol{y}$ is pre-fixed at class 0. As we can see, when conditioned on label 0, almost all the generated samples are really in that class.

As in [19], we investigate whether the models learn to merely copy the data. We visualize the nearest neighbors in the MNIST dataset of several samples generated by CGMMN in terms of Euclidean pixel-wise distance [5] in Fig. 3. As we can see, by this metric, the samples are not merely the copy.

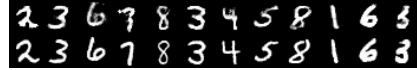

Figure 3: CGMMN samples and their nearest neighbour in MNIST dataset. The first row is our generated samples.

As also discussed in [19], real-world data can be complicated and high-dimensional and autoencoder can be good at representing data in a code space that captures enough statistical information to reliably reconstruct the data. For example, visual data, while represented in a high dimension often exists on a low-dimensional manifold. Thus it is beneficial to combine autoencoders with our CGMMN models to generate more smooth images, in contrast to Fig. 2 where there are some noise in the generated samples. Precisely, we first learn an auto-encoder and produce code representations of the training data, then freeze the auto-encoder weights and learn a CGMMN to minimize the CMMD objective between the generated codes using our CGMMN and the training data codes. The generating results are shown in Fig. 4. Comparing to Fig. 2, the samples are more clear.

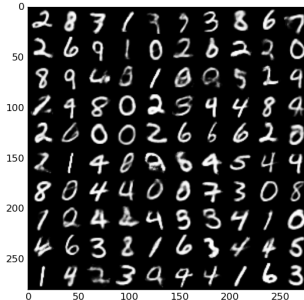

Figure 4: Samples generated by CGMMN+Autoencoder, where the architecture follows from [19].

### 4.2.2 Results on Yale Face dataset

We now show the generating results on the Extended Yale Face dataset [9], which contains $2,414$ grayscale images for 38 individuals of dimension $32 \times 32$. There are about 64 images per subject, one per different facial expression or configuration. A smaller version of the dataset consists of 165 images of 15 individuals and the generating result can be found in Appendix A.4.2.

We adopt the same architecture as the first generating experiment for MNIST, which is a 5-layer MLP with an input dimension of 50 (12 hidden variables and 38 dimensions for conditional variables, i.e., labels) and the middle layer hidden unit number $(64, 256, 256, 512)$. A mini-batch size of $400$ is adopted. The other settings are the same as in the MNIST experiment. The overall generating results are shown in Fig. 5, where we really generate diverse images for different individuals. Again, as shown in Appendix A.4.1, the generated samples are not merely the copy of training data.

### 4.3 Distill Bayesian Models

Our final experiment is to apply CGMMN to distill knowledge from Bayesian models by learning a conditional distribution model for efficient prediction. Specifically, let $\boldsymbol{\theta}$ denote the ran-

dom variables. A Bayesian model first computes the posterior distribution given the training set $\mathcal{D} = \{(\boldsymbol{x}_i, \boldsymbol{y}_i)\}_{i=1}^{N}$ as $p(\boldsymbol{\theta}|\mathcal{D})$. In the prediction stage, given a new input $\boldsymbol{x}$, a response sample $\boldsymbol{y}$ is generated via probability $p(\boldsymbol{y}|\boldsymbol{x}, \mathcal{D}) = \int p(\boldsymbol{y}|\boldsymbol{x}, \boldsymbol{\theta})p(\boldsymbol{\theta}|\mathcal{D})d\boldsymbol{\theta}$. This procedure usually involves a complicated integral thus is time consuming. [15] show that we can learn a relatively simple *student network* to distill knowledge from the *teacher network* (i.e., the Bayesian model) and approximately represent the predictive distribution $p(\boldsymbol{y}|\boldsymbol{x}, \mathcal{D})$ of the teacher network.

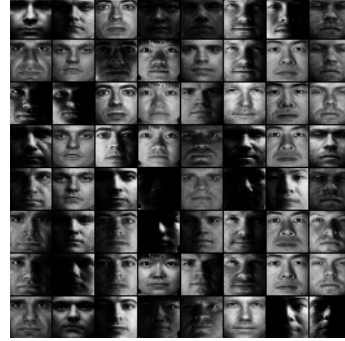

Our CGMMN provides a new solution to build such a student network for Bayesian dark knowledge. To learn CGMMN, we need two datasets to estimate the CMMD objective — one is generated by the teacher network and the other one is generated by CGMMN. The former sampled dataset serves as the training dataset for our CGMMN and the latter one is generated during the training process of it. For high-dimensional data, adopting the same strategy as [15], we sample "near" the training data to generate the former dataset (i.e., perturbing the inputs in the training set slightly before sending to the teacher network to sample $\boldsymbol{y}$).

Due to the space limitation, we test our model on a regression problem on the Boston housing dataset, which was also used in [15, 10], while deferring the other results on a synthetic dataset to Appendix A.3. The dataset consists of 506 data points where each data is of dimension 13. We first train a PBP model [10], which is a scalable method for posterior inference in Bayesian neural networks, as the teacher and then distill it using our CGMMN model. We test whether the distilled model will degrade the prediction performance.

Figure 5: CGMMN generated samples for Extended Yale Face Dataset. Columns are conditioned on different individuals.

We distill the PBP model [10] using an MLP network with three hidden layers and $(100, 50, 50)$ hidden units for middle layers. We draw $N = 3,000$ sample pairs $\{(\boldsymbol{x}_i, y_i)\}_{i=1}^{N}$ from the PBP network, where $\boldsymbol{x}_i$ is the input

Table 3: Distilling results on Boston Housing dataset, the error is measured by RMSE

| PBP prediction | Distilled by CGMMN |
|---|---|
| $2.574 \pm 0.089$ | $2.580 \pm 0.093$ |

variables that serve as conditional variables in our model. For a fair comparison, $\boldsymbol{x}_i$ is generated by adding noise into training data to avoid fitting the testing data directly. We evaluate the prediction performance on the original testing data by root mean square error (RMSE). Table 3 shows the results. We can see that the distilled model does not harm the prediction performance. It is worth mentioning that we are not restricted to distill knowledge from PBP. In fact, any Bayesian models can be distilled using CGMMN.

# 5 Conclusions and Discussions

We present conditional generative moment-matching networks (CGMMN), which is a flexible framework to represent conditional distributions. CGMMN largely extends the ability of previous DGM based on maximum mean discrepancy (MMD) while keeping the training process simple as well, which is done by back-propagation. Experimental results on various tasks, including predictive modeling, data generation and Bayesian dark knowledge, demonstrate competitive performance.

Conditional modeling has been practiced as a natural step towards improving the discriminative ability of a statistical model and/or relaxing unnecessary assumptions of the conditional variables. For deep learning models, sum product networks (SPN) [24] provide exact inference on DGMs and its conditional extension [4] improves the discriminative ability; and the recent work [21] presents a conditional version of the generative adversarial networks (GAN) [5] with wider applicability. Besides, the recent proposed conditional variational autoencoder [28] also works well on structured prediction. Our work fills the research void to significantly improve the applicability of moment-matching networks.

**Acknowledgments**

The work was supported by the National Basic Research Program (973 Program) of China (No. 2013CB329403), National NSF of China Projects (Nos. 61620106010, 61322308, 61332007), the Youth Top-notch Talent Support Program, and the Collaborative Projects with Tencent and Intel.

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
