[Supplementary Material]

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

# A Appendix

## A.1 Gradient Calculation

The CMMD objective can be optimized by gradient descent. Precisely, for any network parameter $\boldsymbol{w}$, we have that:

$$\frac{\partial \hat{\mathcal{L}}^2_{\text{CMMD}}}{\partial \boldsymbol{w}} = \sum_{i=1}^{M} \frac{\partial \hat{\mathcal{L}}^2_{\text{CMMD}}}{\partial \boldsymbol{y}^s_i} \frac{\partial \boldsymbol{y}^s_i}{\partial \boldsymbol{w}},$$

where the term $\dfrac{\partial \boldsymbol{y}^s_i}{\partial \boldsymbol{w}}$ can be calculated via back-propagation throughout the DNN and we use the chain rule to compute

$$\frac{\partial \hat{\mathcal{L}}^2_{\text{CMMD}}}{\partial \boldsymbol{y}^s_i} = \text{Tr}\left(\widetilde{K}_s^{-1} K_s \widetilde{K}_s^{-1} \frac{\partial L_s}{\partial \boldsymbol{y}^s_i}\right) - 2 \cdot \text{Tr}\left(\widetilde{K}_s^{-1} K_{sd} \widetilde{K}_d^{-1} \frac{\partial L_{ds}}{\partial \boldsymbol{y}^s_i}\right).$$

The derivative of the kernel gram matrix (i.e., $\dfrac{\partial L_s}{\partial \boldsymbol{y}^s_i}$ and $\dfrac{\partial L_{ds}}{\partial \boldsymbol{y}^s_i}$) can be calculated directly as long as the kernel function of output samples $\boldsymbol{y}$ is differentiable, e.g., Gaussian RBF kernel $k_\sigma(\boldsymbol{y}, \boldsymbol{y}') = \exp\{-\frac{\|\boldsymbol{y}-\boldsymbol{y}'\|^2}{2\sigma^2}\}$.

## A.2 Implementation details

Here we list some practical considerations to improve the performance of our models.

### A.2.1 Minibatch Training

The CMMD objective and its gradient involve an inverse operation on matrix such as $K_d + \lambda I$, which has $O(N^3)$ time complexity to compute. This is unbearable when the data size is large. Here, we present a minibatch based training algorithm to learn the CGMMN models. Specifically, in each training epoch, we first choose a small subset $\mathcal{B} \subset \mathcal{D}$ and generate an equal number of samples based on the observation $\boldsymbol{x} \in \mathcal{B}$ (i.e., we provide each $\boldsymbol{x} \in \mathcal{B}$ to the generator to get a new sample). The overall algorithm is provided in Alg. 1. To further accelerate the algorithm, we can pre-compute the inverse matrices $\widetilde{K}_d^{-1}$ and $\widetilde{K}_s^{-1}$ as cached data.

Essentially, the algorithm uses a single mini-batch to approximate the whole dataset. When the dataset is "simple" such as MNIST, a mini-batch of size 200 is enough to represent the whole dataset, however, for more complex datasets, larger mini-bath size is needed.

### A.2.2 Finite Case for Conditional Variables

Recall the empirical estimator of conditional kernel embedding operator as mentioned in Sec. 2.3: $\widehat{C}_{Y|X} = \Phi(K + \lambda I)^{-1}\Upsilon^\top$, where we need to compute the inverse of kernel gram matrix of the condition variables. Since the domain of the variables is finite, the gram matrix is not invertible in most cases. Although we can impose a $\lambda$ to make the gram matrix invertible forcibly, this method cannot get the best result in practice. Besides, the main effect of $\lambda$ is serving as regularization to avoid overfitting, not to make the gram matrix invertible [8].

Fortunately, the problem can be avoided by choosing special kernels and estimating the conditional operator directly. More precisely, we use Kronechker Delta kernel on conditioned variables $X$, i.e., $k(x, x') = \delta(x, x')$. Suppose that $x \in \{1, ..., K\}$, then the corresponding feature map $\phi(x)$ is the standard basis of $e_x \in \mathbb{R}^K$. Recall that $C_{Y|X} = C_{YX} C_{XX}^{-1}$, instead of using the estimation before, we now can estimate $C_{XX}^{-1}$ directly since it can be expressed as follows:

$$C_{XX}^{-1} = \begin{pmatrix} P(x=1) & ... & 0 \\ & \ddots & \\ 0 & ... & p(x=K) \end{pmatrix}^{-1}.$$

Obviously, the problem of inverse operator disappears.

### A.2.3 Kernel Choosing

In general, we adopted Gaussian kernels as in GMMN. We also tried the strategy that combines several Gaussian kernels with different bandwidths, but it didn't make noticeable difference.

We tuned the bandwidth on the training set, and found that the bandwidth is appropriate if the distance of two samples (i.e., $\|x - y\|^2/\sigma^2$) is in range $[0, 1]$.

### A.3 Distill Knowledge from Bayesian Models

We evaluate our model on a toy dataset, following the setting in [15]. Specifically, the dataset is generated by random sampling 20 one-dimensional inputs $x$ uniformly in the interval $[-4, 4]$. For each $x$, the response variable $y$ is computed as $y = x^3 + \epsilon$, where $\epsilon \sim \mathcal{N}(0, 9)$.

We first fit the data using probabilistic backpropagation (PBP) [10], which is a scalable method for posterior inference in Bayesian neural networks. Then we use CGMMN with a two-layer MLP architecture, which is of size $(100, 50)$, to distill the knowledge for the PBP network (same architecture as CGMMN) using $3,000$ samples that are generated from it.

(a) PBP prediction        (b) Distilled prediction

Figure 6: Distilling results on toy dataset. (a) is the prediction given by PBP; (b) is the distilled results using our model

Fig. 6 shows the distilled results. We can see that the distilled model is highly similar with the original one, especially on the mean estimation.

### A.4 More Results on Yale Face Dataset

### A.4.1 Interpolation for Extended Yale Face samples

One of the interesting aspects of a deep generative model is that it is possible to directly explore the data manifold. As well as to verify that our CGMMN will not merely copy the training data, we perform linear interpolation on the first dimension of the hidden variables and set the other dimensions to be $0$. Here we use the same settings as in Sec. 4.2.2.

Fig. 7 shows the result. Each column is conditioned on a different individual and we can find that for each individual, as the value of the first dimension varies, the generated samples have the same varying trend in a continuous manner. This result verifies that our model has a good latent representation for the training data and will not merely copy the training dataset.

Figure 7: Linear interpolation for Extended Yale Face Dataset. Columns are conditioned on different individuals.

### A.4.2 Results for smaller version of Yale Face Dataset

Here we show the generating result for the small version of Yale Face Dataset, which consists of 165 figures of 15 individuals. We adopt the same architecture as the generating experiments for MNIST, which is a 5-layer MLP with input dimension 30 (15 hidden variable and 15 dimension for conditional variable) and the middle layer hidden unit number $(64, 256, 256, 512)$. Since the dataset is small, we can run our algorithm with the whole dataset as a mini-batch. The overall results are shown in Fig. 8. We really generate a wide diversity of different individuals. Obviously, our CGMMN will not merely copy the training dataset since each figure of (b) in Fig. 8 is meaningful and unique.

(a) Different individuals      (b) Individual 15

Figure 8: CGMMN generated samples for Yale Face Dataset. Columns in (a) are conditioned on different individuals while the label is 15 in (b).