[Reviews · NeurIPS 2016]

Reviewer 1

Summary

This paper develops a neural network approach paste on a conditional maximum mean discrepancy criterion. the proposed technique is evaluated font standard small benchmark tasks for discriminative and generative capabilities.

Qualitative Assessment

Overall this is an interesting extension for the sub-area of deep learning research focused on building generative models. The paper text is reasonably clear although, like most papers in this area, the math can be a bit difficult to follow at times. The tasks used in experiments are somewhat small but not entirely unreasonable when compared against other recent work in this area. As the algorithm for training the model involves some computationally expensive steps I would ask for some comments in the experiment section on how long the model takes to train compared with existing approaches like DSN. The paper would be much stronger if reference code were made available for the experiments. From the model section a practitioner may have difficulty reproducing the exact models evaluated.

Confidence in this Review

2-Confident (read it all; understood it all reasonably well)


Reviewer 2

Summary

The authors extend moment matching networks to conditional version. Unlike in models that can do update on single example and where the extension is trivial (feed conditional into the network) here it is non-trivial. That’s because we need to match distributions of large number of samples (mini batch) to large number of inputs, but these distributions are not the same anymore (unless same conditional would be used for a given update which is inefficient). The main problem with the paper is that they start with fancy math instead of giving intuition behind this formula and they don't highlight what are the actual computations performed during this algorithm and their efficiency.

Qualitative Assessment

The authors extend moment matching networks to conditional version. Unlike in models that can do update on single example and where the extension is trivial (feed conditional into the network) here it is non-trivial. That’s because we need to match distributions of large number of samples (mini batch) to large number of inputs, but these distributions are not the same anymore (unless same conditional would be used for a given update which is inefficient). The main problem with the paper is that they don’t explain the details behind the formula that is being optimized in the end (eq (2)), how efficient it is, what assumptions are made to make it work so that reader sees what is the actual computation performed. Equally they don’t explain intuition behind it but instead start with formal math. The non-conditional version of the algorithm can be explained (and is in the original paper) without fancy math (e.g. reproducing kernel Hilbert space) - the formula that is minimize is intuitive. It would be good to come up with equally good intuitive explanation for the conditional case. It would be good to answer with the following questions: - How would you intuitively come up with formula (2). - What are the steps from formula (2) to the practical algorithm. Some of it is in the supplement but hard to follow.

Confidence in this Review

1-Less confident (might not have understood significant parts)


Reviewer 3

Summary

This paper proposes a loss function for directed generative models with conditional variables. The method is an extension of generative moment matching networks, which uses an unconditional loss function based on the maximum mean discrepancy criterion. This approach leverages the work on conditional kernel mean embeddings to produce the loss function, and the results are demonstrated on a variety of generative tasks including classification, generating handwitten digits and faces, and distilling knowledge in Bayesian models.

Qualitative Assessment

This is my second time reviewing this paper. The issues the reviewers had last time were minor: why the authors used a low-dimensional space and whether this could scale to a higher-dimensional latent space, the utility of this model compared to others, and other miscellaneous clarifying issues. Looking at the text, these seem to have been answered directly. With respect to Remark 1, it's unclear to me how to map from Equation (2) onto the weighted MMD form. What is the dot notation you're using here? Is it matrix multiplication or the Hadamard product? I would like to see this part given a bit more detail so that the mapping is more obvious. Also the choice of C I think is a bit unfortunate since you're already using C to define the conditional embedding. Theorems 1 and 2 refer to other papers, however Theorem 3 has no reference. A reference should be given to a paper containing a proof, or a proof must be given. The empirical results are decent. In previous reviews it was pointed out that using this model for supervised learning is perhaps not a great application, but the authors stated that it was to highlight the versatility of the model, which I think is fine. The distilled Bayesian inference application is interesting, but it's only used on one dataset with one model so it's unclear whether this will be useful in other cases. I feel like this approach would gain the most traction in pure generative modeling. The experiments in this case are comparable to those of the original GMM paper. The datasets used involve digits and faces, which I think is sufficient to demonstrate that the idea works. In terms of whether others will adopt this method, there would need to be compelling results on more difficult datasets, e.g., CIFAR-10 or LSUN.

Confidence in this Review

3-Expert (read the paper in detail, know the area, quite certain of my opinion)


Reviewer 4

Summary

This paper builds upon the deep generative modeling approach “Generative Moment-Matching Networks” and derives an extension that allows for conditional generation. (This si done by deriving a kernel embedding of the moments defining the conditional distribution in such a way that they can be optimized using the same machinery as the unconditional version.) The proposed model is demonstrated with reasonably competitive performance on predictive modeling, contextual generation, and the distillation of the predictions from a Bayesian model.

Qualitative Assessment

The paper appears to be technically sound and the proposals are well presented. (However, I did not check all the equations in full detail.) In terms of novelty, while the idea of making a conditional extension to the generative moment matching framework is quite natural, the mechanism and derivation of the empirical estimator was not obvious (at least to me). And so in this case, the paper seems to have novelty and usefulness. The derived framework itself should be quite easy to apply, as shown in Algorithm 1, consequently this paper will likely be useful and of relevance to people who are interested in using deep models for generative purposes or to distill more elaborate statistical processes. The paper was well written, clear, and easy to read, and the provided appendix usefully provides more complete information regarding the details of the methods used and for the experiments.

Confidence in this Review

2-Confident (read it all; understood it all reasonably well)


Reviewer 5

Summary

The authors extend the generative moment-matching networks (GMMNs) by considering matching conditional distributions instead of joint distributions. The key idea is to embed the conditional distributions in a reproducing kernel Hilbert space, a technique proposed by Song et al. in 2009. The proposed algorithm CGMMN is evaluated empirically on several real datasets.

Qualitative Assessment

In the definition of $\hat{C}_{Y|X}$ (Line#90), the term related to $\mu_X$ and $\mu_Y$ (more precisely, their estimators) is missing. It is also unusual to assume that $\mu_X$ and $\mu_Y$ are zeroes, i.e., the data is already centered in the RKHS. Since $\hat{C}_{Y|X}$ is at the heart of CGMMN, it is desirable that the authors provide clarifications for this missing term, as well as how to deal with it in gradient computation and real experiments. The proposed approach is new and neat, but the experimental results are somehow disappointing. In terms of predictive performance, on both datasets, CGMMN gives larger error rates than the state-of-the-art algorithm CMMVA. In terms of generative performance, since no comparison between CGMMN and other conditional generative methods (e.g., the conditional generative adversarial nets) is provided, it is unclear whether the proposed algorithm could generate better samples. The paper contains some typos, e.g., “Frobeniu norm” (Line#48), “an universal” (Line#65), and “which the” (Line#238). -----Update after the author rebuttal----- The authors have addressed all my concerns. I will change my scores.

Confidence in this Review

2-Confident (read it all; understood it all reasonably well)


Reviewer 6

Summary

This paper presents conditional generative moment matching networks, an extension of GMMNs to conditional generation / prediction applications. The key to the proposed method is the kernel embedding of conditional distributions and conditional MMD metric for measuring discrepancy of conditional distributions. The proposed CGMMN was tested on classification, conditional generation and distilling Bayesian models.

Qualitative Assessment

The naive approach of extending GMMNs to conditional setting is to estimate a GMMN for each conditional distribution, and all these conditional distributions share parameters through the use of the same neural network. The problem of this approach is that each conditional distribution only has very few examples, and in the case of continuous domain for the conditioning variables, each conditional distribution may only have one single example, causing data sparsity problem. The proposed approach treats all the conditional distributions as a family and tries to match the model with the conditional embedding operator directly rather than matching each individual conditional distributions. The advantage of the proposed approach seems clear, but in some cases I can still see the naive approach do a reasonable job, for example in conditional generation where the conditioning variable takes one of 10 values as in MNIST. It would be interesting to compare to such a naive approach as a baseline. It is nice to see that the conditional MMD objective has a nice formulation, and can be easily computed almost in the same way as the unconditional MMD as pointed out in Remark 1. All the K matrices in equation 2 are independent from the model parameters, and backpropagation can be done in the same way as before. One concern about the proposed approach is that the CMMD objective potentially requires big minibatches, otherwise the stochasticity of the minibatches may dominate the error, on the other hand big minibatches makes the computation of the objective expensive, especially the matrix inverses could be prohibitively expensive to compute. Overall I think the CMMD provides a nice way to model conditional distributions. The experiment results on classification and distillation of Bayesian models are also interesting. The presentation in section 2.3 and 3.1 wasn’t easy to follow. I have to dive into the referenced papers to understand more about what is actually going on. There are a couple minor problems: (1) on line 77, it is mentioned that the MMD estimator is unbiased, which is not true. (2) on line 90, the conditional embedding operator was not consistent with the one given in reference [29]. To get the form used in this paper it seems we need a different definition of the cross covariance operator.

Confidence in this Review

2-Confident (read it all; understood it all reasonably well)